# The Neurobiological Links between Stress and Traumatic Brain Injury: A Review of Research to Date

**DOI:** 10.3390/ijms23179519

**Published:** 2022-08-23

**Authors:** Lexin Zheng, Qiuyu Pang, Heng Xu, Hanmu Guo, Rong Liu, Tao Wang

**Affiliations:** 1Department of Forensic Medicine, School of Basic Medicine and Biological Sciences, Soochow University, Suzhou 215123, China; 2Shanghai Key Lab of Forensic Medicine, Key Lab of Forensic Science, Ministry of Justice, China (Academy of Forensic Science), Shanghai 200063, China

**Keywords:** traumatic brain injury, stress, brain region, neurological dysfunction, biomarker

## Abstract

Neurological dysfunctions commonly occur after mild or moderate traumatic brain injury (TBI). Although most TBI patients recover from such a dysfunction in a short period of time, some present with persistent neurological deficits. Stress is a potential factor that is involved in recovery from neurological dysfunction after TBI. However, there has been limited research on the effects and mechanisms of stress on neurological dysfunctions due to TBI. In this review, we first investigate the effects of TBI and stress on neurological dysfunctions and different brain regions, such as the prefrontal cortex, hippocampus, amygdala, and hypothalamus. We then explore the neurobiological links and mechanisms between stress and TBI. Finally, we summarize the findings related to stress biomarkers and probe the possible diagnostic and therapeutic significance of stress combined with mild or moderate TBI.

## 1. Introduction

More than 50 million people worldwide suffer from traumatic brain injury (TBI) each year and approximately 80% of people will experience one or more mild or moderate TBI event in their lifetime [1,2]. In China, the population-based mortality of TBI is approximately 13 cases per 100,000 people, which is similar to the rates reported in other countries [3]. Although most mild and moderate TBI patients recover in a short period, many experiences persistent neurological dysfunction [4,5]. Although the factors that lead to prolonged neurological symptoms after TBI remain unclear, a history of stress is one key factor that can affect the degree of neurological impairments [4,6]. Stress has diverse effects on various brain regions, including the prefrontal cortex (PFC), hippocampus and amygdala [7,8]. Thus, the adverse consequences of acute and chronic stressors on TBI are worth investigating. The hypothalamic–pituitary–adrenal (HPA) axis and the locus coeruleus–norepinephrine (LC-NE) system also play key roles in stress processing [9], and a further study of these pathways may help to identify relevant stress biomarkers.

In this review, the neurobiological links between neurological dysfunctions and the key brain regions affected by TBI and stress are first reviewed. Next, the effects and mechanisms of stress on neurological dysfunctions after TBI are discussed. Lastly, we aim to identify suitable stress markers and explore the possible diagnostic and therapeutic significance on stress or stress plus mild or moderate TBI.

## 2. Neurobiological Links between TBI and Neurological Dysfunctions

TBI is defined as an alteration in brain structure, or other evidence of brain pathology, caused by an external force [10,11]. From the aspect of distribution of structural damage, we can classify the injury to focal or diffuse [12]. Focal brain injury is caused by the outside forces acting on the skull and resulting in compression of the tissue underneath the cranium at the site of the impact or the tissue opposite to the impact [13]. The location of the impact to the skull determines the cerebral pathology and neurological deficits. By definition, diffuse brain injury is more scattered, and is not linked to a specific focus of destructive tissue damage [14], including widely distributed damage to axons, diffuse vascular injury, hypoxic–ischemic injury, and brain swelling. Regardless of focal or diffuse injury, the characteristic of TBI involves the mechanisms of primary and secondary brain injury [15]. Primary brain injury occurs at the exact moment of insult and results in the disruption of cell plasma membrane [16]. Secondary brain injury occurs after primary brain injury and involves the participation of complicated mechanisms, including excitatory toxicity, mitochondrial dysfunction, oxidative stress, lipid peroxidation, neuroinflammation, and axonal degeneration, and, finally, it induces diverse forms of programmed cell death, such as necroptosis, autophagy, apoptosis, pyroptosis, and ferroptosis [17,18,19,20,21]. TBI induces an increased mitochondrial membrane permeability. Mitochondria trigger a variety of apoptotic signaling pathways via interactions among the bcl-2 family proteins in order to release pro-apoptotic proteins from the intermembrane of mitochondria, which result in apoptosis [22]. Several events following TBI, such as tumor necrosis factor (TNF) release, toll-like receptors (TLR) activation, inflammation, and reactive oxygen species (ROS) production, have the potential to activate necrosis, which involves the upstream assembly of the necroptosome complex formed by the interaction of receptor interacting protein kinase 1 and 3 (RIPK1 and 3) and the downstream RIPK3-mediated phosphorylation of mixed lineage kinase domain-like (MLKL) protein [23], which result in necroptosis. Far more than that, the extent of cell loss following TBI has been correlated with cognitive deficits and long-term prognosis in both clinical and experimental studies [24].

A correct assessment of the degree of injury is essential for the effective treatment of TBI. TBI severity is traditionally determined by several clinical indicators, including the consciousness state, Glasgow coma scale score, presence/duration of retrograde amnesia, and neuroimaging evidence [25,26]. Neurological dysfunction can be divided into physical symptoms and neuropsychiatric symptoms. Neuropsychiatric dysfunctions, including cognitive impairments (executive dysfunction, attention disorder, or memory problems) and emotional/behavioral disorders (depression, anxiety, or sleep disorders) [11,27] are common after TBI, typically lasting 7 to 10 days but sometimes months to years. Therefore, differences in neurological dysfunctions after TBI depend on the severity of brain injury as well as the characteristics of the underlying key brain regions (Figure 1).

### 2.1. TBI and Cognitive Impairments

Forms of cognitive impairments post-TBI range from difficulties with executive function, i.e., attention and problem solving, to deficits in information processing and short- and long-term memory [28,29]. Previous research has shown that the cognitive functions, including memory, attention, and executive function, are resolved within 3 to 6 months after mild or moderate TBI [30], whereas severe TBI can cause cognitive impairments for 6 months or longer [28].

Funahashi [31] describes executive function as a product of the coordinated operation of various processes to accomplish a particular goal in a flexible manner. Executive function is the province of the PFC, and executive dysfunction post-TBI is typically associated with frontal lobe injury. Executive function is mediated by a distributed network guided by the frontal lobe that includes the prefrontal sub-region, posterior cortex, and subcortical structures, such as the basal ganglia and ventral striatum [32]. TBI causes the frontal lobe and subcortical structures, such as the cingulate gyrus, amygdala, striatum, and insula, to be particularly vulnerable [33]. Accordingly, patients with PFC damage show alterations in judgment, organization, planning, and decision making. There are functional and anatomical links between the frontal cortex and striatum [34]. Executive function relies on the efficient operation of cortical striatum circuits, which are often abnormal in cases of executive dysfunction [35,36]. The striatum can be divided into the caudate nucleus, putamen, and nucleus accumbens (NAc) [37]. The disruption of the striatum has been reported in various disorders involving executive dysfunction, such as Huntington’s disease [38], multiple system atrophy (MSA) [39], progressive supranuclear palsy [40], and attention deficit hyperactivity disorder (ADHD) [41]. Working memory is also an important part of executive function. Significant functional changes in the PFC circuit have been shown to reduce the large-scale patterns of brain activity, which is associated with working memory impairment [42,43].

Attention is characterized as selectivity and intensity [44]. Selectivity includes focalized attention with the inhibition of distractors and divided attention, which allows for the performance of two tasks simultaneously [44]. Intensity consists of sustained attention allowing the person to maintain attention levels over prolonged periods of time, and to be kept alert [44]. Impaired attention is one of the most common complaints of TBI survivors [45,46,47]. Based on neuroimaging studies, some psychiatric symptoms have been shown to correspond to functional abnormalities in brain regions, such as selective attention being localized to the anterior cingulate cortex (ACC) [48]. An insufficient activation of ACC can lead to reduced attention to detail and easy distraction. The ACC regulates the activity of the cortex and subcortical regions and influences the ability to control and coordinate their interactions [49,50,51]. Interestingly, Sheth et al. [52] reported resting state functional hyperconnectivity of the ACC in veterans with mild TBI (mTBI).

Several theories about learning and memory propose a stable link between the hippocampus and cortex that strengthens or consolidates memory [53]. Memory impairments caused by TBI may thus be due to alterations in the physiological circuit involving the cortex and hippocampus. The division of memory into short-term and long-term memory remains controversial [54], and memory can be alternatively divided into working, episodic, and semantic memory. The hippocampus is the core of episodic memory and the disruption of the dentate gyrus (DG), CA1, and CA3 regions is thought to be the main cause of episode memory deficits post-TBI [42]. A prior study demonstrated a decreased hippocampal volume in mTBI patients with episodic memory impairment [55]. The integrity of the hippocampus is also important for semantic memory. For example, Klooster et al. [56] reported impoverished semantic memory in patients with hippocampal amnesia. Manns et al. [57] assessed the semantic memory capacity of patients with hippocampus damage and found that semantic memory abilities were impaired—especially anterograde and retrograde memory.

### 2.2. TBI and Emotional/Behavioral Disturbances

The most common chronic emotional/behavioral disturbances that occur after TBI are depression, anxiety, and fear [58]. The amygdala is a key brain region involved in emotional processing [59] and amygdala damage is associated with emotional disorders similar to those that arise after TBI [58,60]. The amygdala also plays an important role in recognizing facial emotion, such as fear, disgust, and anger [61,62].

Depression, which is one of the most common chronic psychoses after TBI [63], is thought to be closely related to changes in the amygdala. Depressed patients often have some negative symptoms, such as sleep disturbance, fatigue (anergia), difficulty with concentration, and anhedonia (apathy) [64]. Studies have shown that depression is associated with increased activation in marginal regions, through which, the amygdala is richly associated with cortical regions [65,66]. The effect of antidepressants is partly via effects on the coupling of the amygdala with other brain regions [67]. In a functional neuroimaging study of patients with diffuse TBI [68], the amygdala was found to process emotions and regulate behavioral and physiological responses to stressors [69]. The PFC, as a significant center of thinking and behavior regulation, is also associated with depression [70]. The PFC can be divided into medial PFC (mPFC) and dorsolateral PFC (dPFC) [71]. Using diffusion tensor tractography, Jang et al. identified dPFC after TBI accompanied by depressive symptoms [72]. The hippocampus is part of the limbic system and has nerve fiber connections with emotion-related brain regions such as the PFC and amygdala. A decrease in hippocampal volume has been observed in patients with TBI [73]. In addition, hippocampal volume is associated with TBI injury severity and neuropsychological function [74].

Anxiety is a common condition in which an anxious mood or state persists without an immediate threat. The amygdala, which plays a key role in regulating anxiety-related behaviors [69], is composed of several parts. Of these, the basolateral amygdala (BLA) and central amygdala (CeA) are particularly important in anxiety management [75,76]. The BLA consists of 80% pyramidal glutamate (Glu) neurons, and 20% γ-aminobutyric acid (GABA) neurons. The CeA, which encompasses the centrolateral (CeL) and centromedial (CeM) nuclei, consists of 95% GABAergic medium spiny neurons [77]. The CeM, which is the main output region of the amygdala, mediates the autonomic and behavioral responses to anxiety via projections to the brainstem [78]. The balance between excitation and inhibition determines the overall degree of amygdala excitability. The hypoactivity of GABAergic neurons and/or an increased activation of glutamatergic neurons leads to amygdala hyperexcitability that manifests as anxiety [79]. Prior research suggested that TBI-induced anxiety-like behaviors were associated with increased glutamatergic neurons and decreased GABAergic neurons within the amygdala [68]. Figueiredo et al. [69] observed that animals exhibited increased anxiety-like behaviors 30 days after TBI.

The amygdala is also key for the acquisition and storage of fearful memory [80]. The BLA is the main region associated with sensory inputs into the amygdala, while the CeM is known as the fear effector structure [81]. The amygdala is also involved in regulating fear-related learning via interactions with other brain regions, such as the cortex and hippocampus. External stimuli information is processed through mechanisms inherent in the amygdala and by interactions with other brain regions to produce fear responses as an output and regulate fear responses [82]. Similarly, Glu receptors and GABA receptors are essential for fear learning and memory [83]. A recent study [84] using single-nucleus RNA sequencing (snRNA-seq) demonstrated a significant increase in Decorin (a small leucinerich proteoglycans) expression in amygdala excitatory neurons after TBI, whereas the knockout of *Decorin* alleviated TBI-related fear conditioning.

Overall, immediate or secondary pathological changes following TBI can lead to abnormal cognitive and emotional function. Through clinical interventions, most non-severe TBI patients recover in a short period without any sequelae. However, some non-severe TBI patients still show delayed and even severe neurological abnormalities [85,86]. Previous studies have found that stress may aggravate or improve neurological dysfunctions following non-severe TBI [87,88].

## 3. Neurobiological Links between Stress and Key Brain Regions

All organisms maintain a complex dynamic equilibrium or homeostasis that is constantly challenged by internal or external stimuli, which are termed stressors [89]. Physiological stress is beneficial to the body in that the body can quickly adapt to changes in internal and external environmental factors. However, pathological stress that is intense and persistent is harmful to the body and can cause physical and mental dysfunction, resulting in many negative adaptation reactions [90]. The brain processes external information and determines the necessary behavior and physiological responses, whether adapting or overloading. As an organ, the brain changes in response to acute and chronic stress [91], and stress hormones can have protective or destructive effects on the brain. Studies have shown that moderate stress facilitated classical conditioning and associative learning [92], in contrast to the chronic stress-induced deficits in spatial and contextual memory and attention [93]. It is noteworthy that different stress paradigms can have a psychological influence or physical impact simultaneously. Psychological stressors may include social order conflicts and competition for resources, as well as restraint and immobilization with accompanying anxiety and fear [94]. Methods of physical stress include, but are not limited to, a lack of food or water, handling, and surgical procedures [95]. The HPA axis and LC-NE system play major roles in these stress responses. Neural circuits between different brain regions, including the hippocampus, amygdala, PFC, and hypothalamus, also play pivotal roles in responding to stress (Figure 1) [96].

### 3.1. Stress and the HPA Axis

The activation of the HPA axis is the primary hormonal response to stress. The initiation of the HPA axis is controlled by corticotropin-releasing hormone (CRH) neurons of the paraventricular nucleus (PVN) [97]. CRH and arginine vasopressin (AVP), which are released by PVN through the pituitary portal to the pituitary gland, act together, acting on the pituitary gland to promote the release of adrenocorticotropic hormone (ACTH) via the circulatory system to the adrenal cortex, thereby promoting the synthesis and release of glucocorticoids (GCs), which act on the body’s organ systems to adapt to changes in the internal and external environment [98]. GCs primarily bind to two receptors in the brain, namely the mineralocorticoid receptor (MR) and GC receptor (GR). The activation of these receptors alters the gene expression profiles slowly and persistently, ultimately affecting brain function [97,99]. A prior study showed that GCs are beneficial for short-term adaptation, but that long-term administration can cause severe damages [100].

### 3.2. Stress and the LC-NE System

The LC in the brainstem contains NE-synthesizing neurons that send diffuse projections throughout the central nervous system (CNS). The LC-NE system plays a major role in behavioral and autonomic responses to stress. During a stressful period, LC-NE neurons supply NE across the CNS to modulate the central stress response [101,102]. NE acts on different adrenal receptors (α_1_, α_2_, and β) and exerts a powerful neuroregulatory function. NE has a higher affinity for α_2_-adrenergic receptors and a lower affinity for α_1_- and β-adrenergic receptors [103]. The LC is involved in the stress response mainly through β receptors located in the BLA [104].

### 3.3. Stress and the PFC

The mPFC is primarily composed of glutamatergic pyramidal neurons (PNs) [105]. The mPFC PNs that orchestrate stress responses are tightly controlled by a complex network of GABAergic interneurons [106]. Although there are several groups of interneurons, the majority of GABAergic interneurons express parvalbumin (PV) and are thus termed PV neurons. Somatostatin (SST) neurons are also thought to regulate the Glu output from the dendritic trees of PNs through synaptic contact [107]. An imbalance between excitatory and inhibitory (E/I) neurotransmission is thought to be the basis for various neuropsychiatric disorders [108]. These interneurons express GRs and have the ability to integrate systemic stress signals. GRs are bound during acute stress, increasing soluble N-ethylmaleimide-sensitive factor attachment protein receptor (SNARE) protein complexes (which mediate synaptic vesicles to fuse with the anterior membrane) in the presynaptic membrane [109,110]. Therefore, acute stress increases the excitability of glutamatergic neurons, as shown by extracellular Glu, postsynaptic membrane N-methyl d-aspartate receptor (NMDAR), and alpha-amino-3-hydroxy-5-methyl-4-isoxazole-propionic acid receptor (AMPAR) expressions, and increases in NMDA and AMPA-mediated excitatory currents. Animals studies have shown that acute exposure to stress or the administration of GCs increased the Glu release from the PFC [110,111]. Using microdialysis, it has been shown that exposing rats to tail-pinch, restraint, or forced-swim stress induces a marked, transient increase in extracellular Glu levels in the PFC [107]. Conversely, repeated stress has been shown to inhibit Glu delivery in the PFC by promoting the degradation of Glu receptors in juvenile rats [111].

The PFC has extensive neuronal connections with other brain regions that regulate behavior, cognition, and emotions [112]. Catecholaminergic neurons projections to the cerebral cortex stem from two main sources, namely NE neurons of the LC in the brainstem and dopamine (DA) neurons of the ventral tegmental area (VTA) in the midbrain. The PFC is a main cortical target of both NE and DA innervations [113]. NE and DA each have an ‘inverted U’-shaped influence on working memory, such that either too little or too much impairs PFC function [114,115]. A previous study showed that α_1_-adrenoceptor stimulation in the PFC contributes to stress-induced cognitive impairments [116]. The low NE level present under control (non-stress) conditions optimizes working memory by engaging α_2A_-receptors, whereas the high NE level during stress impairs PFC function by stimulating lower-affinity α_1_-receptors and β_1_-receptors [117]. An excessive activation or blocking of dopamine D1 receptors (D1Rs) during working memory stress can both lead to working memory deficits [114]. Memory deficits due to the excessive activation of D1R can be prevented by D1R antagonists [118], whereas spatial working memory deficits mediated by an increased D1R density are improved by D1R agonists [119]. Under normal circumstances, the extensive connections of PFC coordinate brain activity and regulate the catecholamine input.

### 3.4. Stress and Hippocampus

The hippocampus is rich in GRs and MRs [120], making it a key regulatory region of the HPA axis. Stress has a great impact on excitatory transmission and the synaptic plasticity of the hippocampus [121]. Excitatory amino acids and NMDAR play important roles in episodic memory function, which is dominated by the hippocampus. Excitatory amino acids produce long-term potentiation (LTP) on synapses [122]. Thus, the plasticity of synaptic connections in the hippocampus and LTP are the basis of learning and memory. LTP production relies on synaptic connections between cells in the CA1 and CA3 regions of the hippocampus, which, in turn, depend on Glu as a neurotransmitter [123]. During stress, GCs increase and stimulate Glu release from the hippocampus, which, in turn, inhibits DG proliferation [124]. Studies have shown that exposure to predator odor (2,4,5-trimethythiazole, TMT) causes a stress response in rats, as demonstrated by elevated adrenal steroid levels, DG excitation, and the rapid inhibition of DG proliferation [125]. CA3 dendritic atrophy is suppressed when an excitatory input pathway is damaged. Antagonism with NMDAR can inhibit stress-induced CA3 dendritic atrophy [126]. Interestingly, morphological damage of the CA3 region was reversed within 21 days in a rat model after the end of chronic stress [127]. Chronic stress can also produce CA1 apical dendritic retraction, although stressors tend to be more severe than what is needed to produce an apical dendritic retraction in the CA3 region [128].

### 3.5. Stress and Amygdala

The amygdala plays a key role in physiological and behavioral responses to stress and is characterized by high inhibitory tension mediated by GABA at rest. Stress causes hyperactivity of the amygdala, which is often accompanied by a reduction in inhibition controls [129]. Under physiological conditions, mPFC exerts top-down inhibitory control over amygdala activity, limiting its output and thereby preventing an inappropriate expression of emotions. Under stress conditions, the amygdala activates stress pathways in the hypothalamus and brainstem to induce high levels of NE and DA release, thereby impairing PFC regulation but strengthening amygdala function [130]. In such cases, the PFC control of stress becomes defective, resulting in aberrant amygdala activation and deficits in emotion and behavior [131]. Thus, during stress, the orchestration of the brain’s response patterns switches from the slow, thoughtful PFC regulation to the reflexive, rapid emotional regulation mediated by the amygdala and related subcortical structures [11].

Stress causes the remodeling of amygdala neuronal projections [132]. In contrast to stress-induced dendritic retraction seen in the hippocampus, projecting neurons within the BLA showed persistent dendritic hypertrophy after chronic stress, but dendritic contractions after acute stress [7]. Thus, the morphology of the amygdala and hippocampus showed opposite adjustments after chronic stress. In contrast, Glu is enhanced in both the amygdala and hippocampus after stress. The increase in Glu levels activates NMDAR in BLA, thereby delaying the increase in synaptic spines. A prior study showed that the GR agonist dexamethasone (DEX) enhanced fear resolution via a dose-dependent regulation of the methylation of the GR partner FK506-binding protein 5 (FKBP5) in the BLA [133]. The amygdala is also a major extrahypothalamic source of corticotropin releasing factor (CRF)-containing neurons and has high expression levels of the two cognate CRF receptors. During chronic stress, the repeated activation of CRF receptors leads to an increased NMDAR-mediated Ca^2+^ inflow [134], which inhibits the polymerization of tubulin dimers responsible for microtubule and neurite elongation. If Ca^2+^ is sustained at high levels, microtubules and microfilaments will be depolymerized to trigger dendritic regression [135].

Thus, under stress conditions, the HPA axis and LC-NE systems act with central GRs through the final metabolites GCs and NE. The activation of MRs and adrenal receptors affects the balance of excitatory and inhibitory neurons in the PFC, hippocampus, and amygdala, as well as neuronal plasticity, ultimately affecting working memory, emotions, and other neurological functions. Taken together, these findings indicate that both stress and TBI can lead to deficits in the corresponding brain regions.

## 4. Neurobiological Links between Stress and TBI

Patients recovering from non-severe TBI often experience increased sensitivity to physical and/or psychological post-injury stressors, which may result in significant brain damages and neurological impairments [136,137]. Early life stress (ELS), postnatal stress, or pre-injury stress may also affect the neurological, cognitive, and affective sequelae after a non-severe TBI in adolescence and adulthood [138]. In addition, TBI and post-trauma brain injury (PTSD) are most often strongly connected. Several studies have reported neurological links between stress and TBI and developed models of their interactions. The following section describe these studies (Table 1).

### 4.1. TBI and Post-Injury Stress

Restraint is commonly described as restricting the range of locomotion, but does not attempt to limit specific limb movement [95]. The attraction for using these types of restraint may be due to the fact that they rarely, if ever, involve any bodily harm to the animal subject once the period of the restraint is terminated [95]. This ensures that any long-term effects of stress observed are due to the stressor that was applied, rather than to the physical repercussions of an irreversible or chronic injury [95]. Griesbach et al. [139] utilized a mild fluid-percussion injury (mFPI) plus a post-traumatic stress model to examine the effects of stress on neuroplasticity post-TBI. An increase in GR and a decrease in pro-brain-derived neurotrophic factor (BDNF) in the hippocampus were observed in the FPI plus restraint stress (RS) or DEX groups relative to the control group [139]. Rowe et al. [140] showed that rats with mild and moderate TBI induced by FPI had elevated resting plasma corticosterone levels at 6 and 24 h post-injury, but lower resting plasma corticosterone levels at 7, 14, 28, and 54 days post-injury. Further research [140] demonstrated that, independent of TBI severity, RS increased plasma corticosterone levels at acute stage post-stressor initiation at 7, 14, 28, and 54 days post-injury. Moreover, as the RS duration increased, plasma corticosterone release in the TBI group gradually decreased [140]. Interestingly, compared with low-dose DEX, a high dose of DEX lowered corticosterone levels post-mild and moderate TBI [140]. Our previous study [141] using a moderate controlled cortical injury (CCI)-induced TBI plus RS model showed that RS significantly reduced body weight recovery and delayed the restoration of neurological functions, including motor function, cognitive function, and anxiety-like behaviors, and exacerbated brain lesion volume post-moderate TBI. Our experiment also provided evidence that RS worsened cell damage and BBB leakage and resulted in over-activated endoplasmic reticulum stress (ERS)-mediated neurodegeneration, apoptosis, and autophagy in the injured cortex after moderate TBI [141]. Thus, monitoring and regulating stress levels in patients after non-severe TBI can support recovery.

Algamal et al. [142] used a repeated unpredictable stress (RUS) in combination with repetitive mild TBI (r-mTBI) to assess the impact of repeated stress exposures on TBI. The RUS procedures involved unstable social housing with a different mouse every day, unpredictable exposure to predator odor (TMT) under RS, and inescapable foot shock. Animals receiving r-mTBI (×5) were exposed to a single closed-head injury 1 h after each foot shock. RUS-alone mice showed severe weight loss, traumatic memory impairment, and anxiety-like and passive stress-coping behaviors relative to control mice; however, combined r-mTBI and RUS resulted in an apparent amelioration of stress-related behaviors compared with r-mTBI alone [142]. Molecular studies have found that RUS reduced the dendritic spine GluN2A/GluN2B ratio and pro-BDNF level in the hippocampus and augmented astrogliosis in the corpus callosum induced by r-mTBI [142]. Thus, mild post-injury stress may improve neurological dysfunction post-mild TBI.

Klemenhagen et al. [143] used a mouse model of repetitive concussive TBI (rcTBI, two closed-skull blunt impacts 24 h apart) plus post-injury foot shock stress to examine the impact of stress on TBI. TBI plus stress mice had severe cognitive impairments and increased depression-like behaviors compared with TBI alone mice [143]. In another study using a rat model of mTBI and post-injury repeated immobilization and tail-shock stress [144], the rats showed symptoms of anxiety and memory impairments, as well as an abnormal expression of the mitochondrial electron transport chain (ETC) complex and pyruvate dehydrogenase (PDH) in the hippocampus relative to the mTBI alone group [144]. In another study, stressed injured model animals [145] established by blast-induced TBI (bTBI) and post-injury repetitive unpredictable stressors showed increased anxiety-like behaviors that returned to normal at 2 months and significant spatial memory impairments lasting up to 2 months, relative to bTBI alone. Moreover, compared to the bTBI group, serum levels of corticosterone, creatine kinase (CK)-BB, neurofilaments-heavy (NF-H), neuron-specific enolase (NSE), glial fibrillary acidic protein (GFAP), and vascular endothelial growth factor (VEGF) in the hippocampus and PFC were elevated in animals exposure to bTBI and post-injury stress [145]. Thus, repeated stress may contribute to long-term neurological and neuropsychiatric morbidity post-TBI.

The lifestyle following injury, including differential social interactions, may modulate the extent of secondary injury following TBI [146]. Doulames et al. [146] utilize closed head injury (CHI) acts in mice that were either housed in isolation or with their original cagemates (‘socially housed’) for 4 weeks. The study showed that CHI impaired novel object recognition (NOR) in both isolated and social mice [146]. However, Y maze navigation was impaired in isolated but not social mice. In addition, CHI increased serum corticosterone in both isolated and social mice, which was exacerbated by isolation. Interesting, a dominance hierarchy was shown in socially housed mice, in which, the most submissive mouse displayed indices of stress that were identical to those observed for isolated mice [146]. Khodaie et al. [147] explored the impact of post-injury social isolation on the anatomical and functional deficits post-TBI in male rats. The TBI rats with social isolation showed deteriorative memory impairments and increased numbers of dark neurons, apoptotic cells, and caspase-3 positive cells in the hippocampal CA3 region compared to TBI alone rats [147]. This study demonstrated the harmful effect of social isolation on anatomical and functional deficits induced by TBI, suggesting that the prevention of social isolation may improve TBI outcomes. These findings showed that social interaction may impact the recovery of TBI.

Griesbach et al. [148] constructed a rat model of FPI and found that forced, but not voluntary, running wheel exercise led to elevated plasma corticosterone and ACTH levels on days 1, 4, 7, and 14 compared with FPI alone. Therefore, forced exercise with a strong stress response may not be beneficial during the early post-injury period. Tapp et al. [149] examined the effects of transient mechanical sleep fragmentation (SF) in the subacute and chronic stage following FPI. They found that post-TBI SF increased the cortical expression of stress-associated genes characterized by the inhibition of the upstream regulator *NR3C1*, which encodes GR, whereas TBI plus SF enhanced cortical microgliosis and increased the expression of pro-inflammatory glial signaling genes characterized by the persistent inhibition of *NR3C1* [149]. In addition, post-TBI SF increased neuronal activity in the hippocampus and suppressed activity in the hypothalamic PVN. These findings demonstrated that post-injury SF engages the dysfunctional post-injury HPA-axis, enhances inflammation, and compromises hippocampal function.

In the research of stress and TBI, PTSD has been a topic of long-standing interest. PTSD was considered as a type of extreme stress [157]. PTSD is the simultaneous presentation of four symptom clusters—intrusions, avoidance, negative cognitions, and hyperarousal—initiated after the experience of a traumatic event [158]. Although PTSD is not a requisite sequela of TBI, TBI and PTSD are often comorbid [159]. In a prior study [160], male soldiers were examined before deployment to Afghanistan and at a 12-month post-deployment follow-up. The results [160] showed that lower hair cortisol concentrations (HCC) and lower cortisol stress reactivity (measured by the Trier Social Stress Test) were predictive of a greater increase in PTSD symptomatology in soldiers who had experienced new-onset traumatic events. Therefore, attenuated cortisol secretion may be a risk marker for the subsequent development of PTSD symptomatology upon trauma exposure [160]. Another research [150] completed blood biomarker determinations in a total of 230 veterans who were deployed to Iraq and/or Afghanistan within the last 10 years. The result showed the reduced serum CRF level in PTSD participants, relative to TBI and healthy controls, in Iraq and Afghanistan veteran cohorts. Zadeh et al. [151] developed a mouse model of co-morbid TBI/PTSD by combining the CHI model with the chronic variable stress (CVS) model. Compared to CVS or CHI alone, CVS plus CHI increased the microglia number in the DG, CA1, and CA3 and produced greater behavioral impairments and neuroinflammation [151]. From the above-mentioned studies, it appears that TBI plus PTSD may cause the HPA axis disorder and increased inflammation. However, it is far more than that: physical symptoms in different categories often overlap in PTSD, such as neuroendocrine disorder (HPA axis disorder, abnormalities in thyroid, altered sympathetic function), and decrease in immunity [157]. Therefore, molecular studies involved in these pathways have the potential to serve as potential diagnostic biomarkers.

Overall, these studies suggest that persistent, severe post-injury stress aggravates neurological dysfunctions post-TBI; however, mild post-injury stress may improve neurological deficits post-TBI.

### 4.2. TBI and Pre-Injury Stress

In order to survive, an organism must be able to adapt to its environment. These adaptations occur during critical windows of development because of genetics and epigenetics. This concept is named programming theory [161]. If prenatal or early life environmental conditions do not match those later in life, this programming may be maladaptive and predispose individuals to disease [161]. Therefore, early life stress factors are crucial in explaining the different outcomes after TBI. Maternal separation, a classic model of ELS, is commonly used to establish models of acute or repetitive stress as it can lead to malnutrition and hypothermia in pups [152]. Sanchez et al. [152] established a maternal separation plus TBI animal model to study the effects of ELS on TBI. ELS was induced by separating Sprague Dawley rat pups from their nursing mothers for 3 h daily on post-natal days 2–14, and male animals received mild to moderate FPI at 2 months of age [152]. Behavioral testing [152] found that the combination of ELS with TBI in adulthood impaired hippocampal-dependent learning, as assessed by the contextual fear conditioning, water maze task, and spatial working memory. In addition, ELS plus TBI resulted in more severe cortical atrophy and hormonal stress response relative to TBI alone. Thus, ELS is a risk factor that can worsen post-TBI outcomes. In a rat model of ELS plus pediatric mTBI induced by mild CCI [153], ELS preceding mTBI did not worsen executive function, as shown by the attentional set-shifting test (AST), but did result in increased hippocampal IL-1β relative to mTBI alone. Another study using an ELS (maternal separation) plus pediatric mTBI [154] model showed that this combination led to decreased spatial learning and memory deficits and increased microglial activation in the area adjacent to the injury and the contralateral CA1 hippocampal subfield compared to the sham group [154]. Thus, ELS is a risk factor for cognitive impairments following mild to moderate TBI occurring in adolescence and adulthood.

Using a TBI and pre-injury foot shock stress mouse model, Klein et al. [155] found that TBI or foot-shock stress individually resulted in a significant increase in membrane excitability and spontaneous excitatory postsynaptic currents (sEPSCs) in lateral amygdala pyramidal-like neurons. However, pre-injury stress and TBI led to weakened sEPSC activity compared with either condition alone [155]. The authors speculated that stress and TBI may lead to the hyperexcitation of the amygdala through various mechanisms and that these pathways counterbalance each other in cases of combined injury.

Davies et al. [156] established a rat model combining mTBI and pre-injury social stress. Rats that suffered from social defeat stress or mTBI alone experienced greater anxiety-like behaviors, as shown by the decreased time spent in the open arms of the elevated plus maze (EPM) relative to the control group [156]. However, this effects decreased with the combination of social defeat stress plus mTBI. Moreover, rats exposed to both social defeat and mTBI exhibited greater impaired contextual fear extinction compared with social defeat stress or mTBI only rats [156]. A study of the underlying mechanism found that the serotonergic system, including 5-hydroxytryptophan (5-HTP) and 5-hydroxyindoleacetic acid (5-HIAA) in the hippocampus and amygdala, as well as DA in the dorsal hippocampus and NE in the amygdala, were significantly higher in the social defeat plus mTBI group than the social defeat stress or mTBI alone group [156].

In these limited studies on the effects of stress on TBI, neurological functions, including cognition, and emotion, were the key focus. In addition, blood metabolism markers and protein expressions in key brain regions post-stress have become the focus of experimental research.

## 5. The Biomarkers of Stress

In addition to neurological examination, biomarkers detection is essential in most animal stress models [139,160]. Since the HPA axis and LC-NE system play major roles in the stress response, we focused on neuroendocrine factors that are potential biomarkers to evaluate the physiopathological process of the stress response. We also reviewed changes in metabolites of other physiopathological pathways, such as neurotrophic factors, neurotransmitters, inflammation factors, and oxidative stress after stress induction, and explored their diagnostic and therapeutic values as biomarkers of stress (Figure 2).

### 5.1. Stress and Neuroendocrine Factors

The HPA axis plays a very important role in stress, and chronically stressed patients usually have HPA axis disorders [162]. The hormones critically involved in each step on the HPA axis are thus potential markers that could be used to diagnose or determine the prognosis of stress.

#### 5.1.1. GCs

Under stressful conditions, the HPA axis increases the GC content in blood and exerts its effect on the targeted organ via binding and activating the MRs and GRs [9]. In the case of a low GC concentration or stress, MRs are preferentially occupied due to their strong affinity with GC [163]. As the GC concentration increases, low-affinity GRs are occupied and gradually activated, eventually leading to the termination of the stress response [163]. There is an equilibrium between MRs and GRs; when the balance is disturbed, the body is unable to adapt to stress and experiences dyshomeostasis [164]. Therefore, the measurement of GCs in blood and MR and/or GR expressions in related tissues may be beneficial for evaluating stress responses. Previous studies [165] have quantified GC in hair and have shown that hair cortisol is an effective biomarker of human psychosocial stress [166], suggesting that it may be a reliable biomarker of long-term HPA axis activity.

#### 5.1.2. CRF

CRF responds to stress in the hypothalamic PVN and is considered the initial activator of the HPA axis [167]. Notably, anxiety-like behaviors in rats are accompanied by decreased CRF expression in the PVN of the hypothalamus and the DG of the hippocampus after chronic RS (CRS), whereas the expression of CRF in the PVN and DG increased after exposure to acute stress [168]. The presence of a monoclonal antibody targeting CRF has been shown to inhibit the HPA axis and reverse the stress-induced behavioral deficits in a mouse model of chronic stress [169].

### 5.2. Stress and Neurotrophic Factors

There is a wealth of evidence that stress affects the expression of neurotrophic factors and that some neuroprotectants function to enhance neurotrophic factors and regulate neuroplasticity in animal stress models [170]. As neurotrophic factors are found in both the brain and peripheral blood, they are suitable as biomarkers for stress-induced psychiatric disorders [171].

#### 5.2.1. BDNF

At present, BDNF is the most widely studied neurotrophic factor in stress and psychiatric research [172,173,174]. BDNF is secreted by various CNS cells, including neurons and astrocytes, and can pass through the blood–brain barrier (BBB) [175]. Thus, blood BDNF reflects central BDNF [176,177], which makes it a reliable peripheral biomarker for brain activities. The diverse effects of BDNF at excitatory synapses are mainly determined by the activation of tropomyosin-related kinase receptor (TrkB) and downstream signaling pathways [178]. As a key neurotrophic factor regulating synaptic plasticity, neurogenesis, and behavioral outcomes, BDNF has been studied in several stress-related behavioral paradigms [179]. In stress models, BDNF mRNA and protein levels are decreased in the PFC and hippocampus [180,181,182] whereas BDNF expression is increased in the amygdala [183], which is accompanied by depression-like and anxiety-like behaviors. However, other studies have reported the opposite expression of BDNF mRNA in the above brain regions after stress [184,185]. We speculate that differences in BDNF expression are due to diversity in the stress models and individual heterogeneity.

#### 5.2.2. Vascular Endothelial Growth Factor (VEGF)

VEGF, which is the main growth factor responsible for angiogenesis, can enhance neuronal proliferation in the hippocampus [186]. A prior study found that chronic stress reduced cell proliferation and the expression of VEGF in the DG of hippocampus in adult rats [187]. In contrast, another study found that the expression of VEGF and its receptor VEGFR-2 in the PFC increased under chronic stress [188]. The neuronal density in the CA1 and DG and VEGF in the hippocampus are elevated after acute foot shock [189]. As the type and duration of stress seem to have different effects on the expression of VEGF in different brain regions, further research is warranted.

### 5.3. Stress and Neurotransmitters

Responses to a stressful situation involve the activation of neurotransmitter systems—namely, DA, NE, Glu, and GABA. Notably, these systems are also linked to emotional disorders.

#### 5.3.1. DA

DA is the main catecholamine produced in the substantia nigra (SN) and VTA [190,191]. Dopaminergic neurons in the SN and VTA project to different brain regions, such as striatum, NAc, and PFC. DA release and metabolism, particularly in the limbic system of the midbrain, change with stress stimulation, and the enhancement or inhibition of DA release may be related to the intensity and duration of stress [192]. Rodents exposed to acute restraint and fixed stress (10–240 min) showed an immediate increase in the DA level in NAc and mPFC [193]. In addition, acute foot shock stress (10–30 min) led to increased extracellular DA in the NAc and mPFC. Other stressors, such as tail pinching, short-term handling, and psychological stress, have also been shown to increase DA levels in the NAc and mPFC [193]. Decreased D1R in the NAc and PFC has been reported in a mouse CRS model that also showed depressive-like behaviors [194]. Another study noted that exposure to unavoidable stress for 3 weeks reduced DA in the NAc [195].

#### 5.3.2. NE

Abnormalities in the synthesis and metabolism of monominergic neurotransmitters (DA and NE) are thought to be associated with depression [196]. Central NE is primarily synthesized and secreted by sympathetic and norepinephrinergic neurons. Under the RS [197], the biosynthesis of catecholamine in the adrenal medulla of rats increased, and the effective participation of the adrenal glands maintained the homeostasis. NE is involved in anxiety-like behaviors induced by acute stress, whereas norepinephrinergic derivatives mediate the development of persistent behaviors after stress [198]. Of note, a significant decrease in NE and its metabolites has been reported in a CRS model [199]. During RS, NE levels decrease in the hippocampus and cortex, which are associated with behavioral changes [200]. Several studies have reported increased NE in peripheral blood after stress [201,202,203]. Overall, NE shows the opposite expression in peripheral blood and the CNS, which may be related to its different origins.

#### 5.3.3. Glu and GABA

Glu is the main excitatory neurotransmitter in the CNS and is abundant in the frontal cortex and hippocampus, whereas GABA is the main inhibitory neurotransmitter [204]. In the synaptic cleft, GABA is sensed by two types of receptors, namely GABA_A_ and GABA_B_ [205]. GABA_A_ receptors are ligand-gated ion channels that mediate fast responses by counteracting potentials by increasing the chloride ion permeability of the neuronal membrane [205]. Autopsy studies and serum and cerebrospinal fluid analyses suggest that imbalances between Glu and GABA levels may be related to the pathophysiology of depression [206,207]. A prior study demonstrated that exposure to acute stress or GCs rapidly increased the Glu release in brain regions such as the hippocampus [208]. Furthermore, neuroelectrophysiological recordings have demonstrated that both NMDAR and AMPAR-mediated synaptic currents are significantly increased in PFC pyramidal neurons under acute stress models [209]. Moreover, a decrease in the GABA release and its transporters-related gene and protein in the mPFC has been found in chronic unpredictable mild stress (CUMS)-induced depression model mice [210].

### 5.4. Stress and Inflammatory Factors

The effect of stress on the inflammatory environment is a key mechanism by which stress can affect health [211]. Clinically, GCs are known for their immunosuppressive and anti-inflammatory properties. However, some studies have demonstrated that GC also has pro-inflammatory effects [212,213]. Pro-inflammatory and anti-inflammatory mechanisms seem to depend on the type and intensity of the stressors. Of the inflammatory factors, interleukin-6 (IL-6) and C-reactive protein (CRP) have been shown to play a pro-inflammatory role under stress [214].

#### 5.4.1. IL-6

Yang et al. [199,215] reported increased IL-6 in the peripheral blood and the hypothalamus in a mouse RS model. Stress selectively activates neuron-specific expressing IL-6 in the hypothalamus, which may form the basis of neurobiological links between stress and the inflammatory response [216]. Both repeated social defeat stress (RSDS) and CRS have been shown to induce increases in IL-6 mRNA in the blood and brain [217]. In addition, increased IL-6 is related to depressive-like behaviors under chronic social stress [218].

#### 5.4.2. CRP

CRP is secreted into the blood during the inflammation process, mainly in response to IL-6 signaling. CRP has been shown to increase significantly in response to stressful events [219]. When volunteers were deprived of sleep, their circadian rhythms were interrupted and the blood CRP increased significantly [220]. In a CUMS model, animals showed a significant reduction in neurons in PFC and the hippocampus, accompanied by elevated serum levels of cortisol and CRP [221]. Thus, stress can increase CRP levels in the blood, whether due to acute or chronic stress.

### 5.5. Stress and Oxidative Stress

The brain, especially the hippocampus, cortex, and amygdala, is susceptible to oxidative stress damage [222,223]. Psychosocial stressors can induce ROS and lipid peroxidation products such as malondialdehyde (MDA).

#### 5.5.1. ROS

When an organism experiences stress, large amounts of ROS are released, resulting in an imbalance between the oxidative and antioxidant systems, ultimately leading to oxidative stress [224,225]. In studies using common models of psychosocial stress such as RS and social isolation, ROS in the hippocampus, PFC, and serum are reported to be significantly increased [226,227,228,229]. Thus, excessive ROS production is the main pathological factor in the stress-induced hippocampal injury. The application of antioxidants can relieve hippocampal oxidative stress by inhibiting ROS, thereby improving chronic stress-induced hippocampal damage as well as learning and memory dysfunctions [226]. Exogenous antioxidants can also enhance resistance to stress and mitigate the negative consequences of stress through various pathways [230,231].

#### 5.5.2. MDA

Lipid peroxidation is caused by the action of ROS on lipids, such as cell membrane lipids [232,233]. Early lipid peroxidation is reflected as a higher lipid ROS level, whereas MDA is more likely to reflect a lower lipid peroxidation level. Mice exposed to CRS have been shown to develop significant memory impairment and anxiety-like behaviors, which led to increased lipid peroxidation in the brain and serum, manifested by a significant increase in MDA levels. After treatment with antioxidants, anxiety-like behaviors were significantly improved and lipid peroxidation in the serum and brain was reduced [234,235]. Studies have also shown that lipid peroxidation is associated with the severity of depression [236].

To examine the effects of stress on TBI, blood-derived biomarkers are urgently needed. In 2018, the FDA authorized the use of a blood test for GFAP and ubiquitin carboxy-terminal hydrolase L1 (UCH-L1) in mTBI [237]. The investigation of a range of astroglial and neuronal biomarkers, including calcium-binding protein (S100B), GFAP, and UCH-L1, aims at improving the accuracy of TBI diagnosis and the associated decision-making process [238].

Since stress involves multiple physiopathological metabolic pathways, screening for stress biomarkers is not that difficult. However, the same biomarker may be expressed differently in diverse stress models and the expression of the same biomarker in central and peripheral blood may differ. Therefore, researchers should pay attention to the heterogeneity of stress models and biomarkers. The discovery of stress biomarkers could help us to find therapeutic targets, while TBI biomarkers help us to monitor prognosis.

## 6. Conclusions and Expectation

TBI patients often present with unconsciousness and memory loss during the acute stage post-injury, and may suffer cognitive, emotional, and functional impairment in the subacute and recovery stage. The pattern and extent of TBI depend on the location and duration of the injury, as well as other confounding factors, such as childhood distress, family factors, steroid use, depression, anxiety, and life stress. Numerous studies have shown that stress can lead to cognitive disturbances such as depression and anxiety via effects on key brain regions. Therefore, there may be some neurobiological links between stress and TBI that lead to altered stress responses, ultimately aggravating or improving neurological dysfunction after TBI. Based on the existing findings, the treatment of neurological dysfunction after TBI may involve monitoring and regulating stress.

Neuroimaging analyses of TBI patients usually involve CT, MRI, diffusion tensor imaging (DTI), susceptibility weighted imaging (SWI), and resting-state functional MRI (fMRI). These analyses examine the physiopathology of TBI but can also assess certain brain functions, such as perception and cognitive tasks, which include memory and concentration. Blood biomarkers can play an important role in the clinic practice. Based on the stress response involving the HPA axis and the LC-NE system, the diagnosis and prognosis of the stress state can be assessed by detecting levels of these biomarkers.

In the existing studies on the combination of stress and TBI, the results often focus on changes in cognitive and emotional function in animals. The determination of GCs in the blood has also received attention. Because GRs are present in multiple brain regions, the functions governed by these regions may be mainly regulated by GCs. Therefore, the determination of GC content (central or peripheral) and the expression of GRs in the brain should be studied in animal models of stress plus TBI. The clearance of excess GCs or blocking of GRs may be potential treatment strategies for TBI patients.

The development of stress or TBI alone or stress plus TBI is not a single physiopathological phenomenon but a complex disease process. The discovery of appropriate biomarkers will not only support the diagnosis of the stress state before and after TBI but will also provide intuitive and objective indicators for improving neurological dysfunction. Glu/GABA plays a crucial role in memory impairments and emotional disturbances. The detection of Glu/GABA allows us to better understand the tendency of neurological changes, while the Glu/GABA receptor is a potential therapeutic target in stress plus non-severe TBI patients. Since GCs, NE, and DA act on the corresponding receptors in the brain to impact the expression of Glu/GABA, the receptors may also be therapeutic targets for non-severe TBI patients.

In summary, there are a series of neurobiological links between stress and TBI and severe stress mediating the adverse effects of TBI. Therefore, monitoring stress levels by detecting the biomarkers in patients recovering from non-severe TBI warrants consideration in the future.

## Figures and Tables

**Figure 1 ijms-23-09519-f001:**
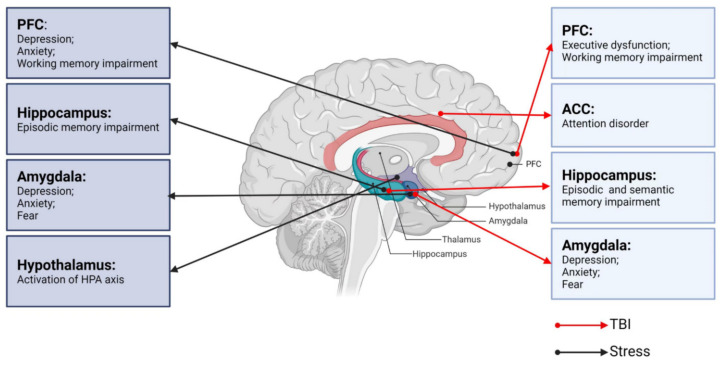
Neurobiological links between TBI, stress, and neurological dysfunctions. PFC: prefrontal cortex; ACC: anterior cingulate cortex; TBI: traumatic brain injury.

**Figure 2 ijms-23-09519-f002:**
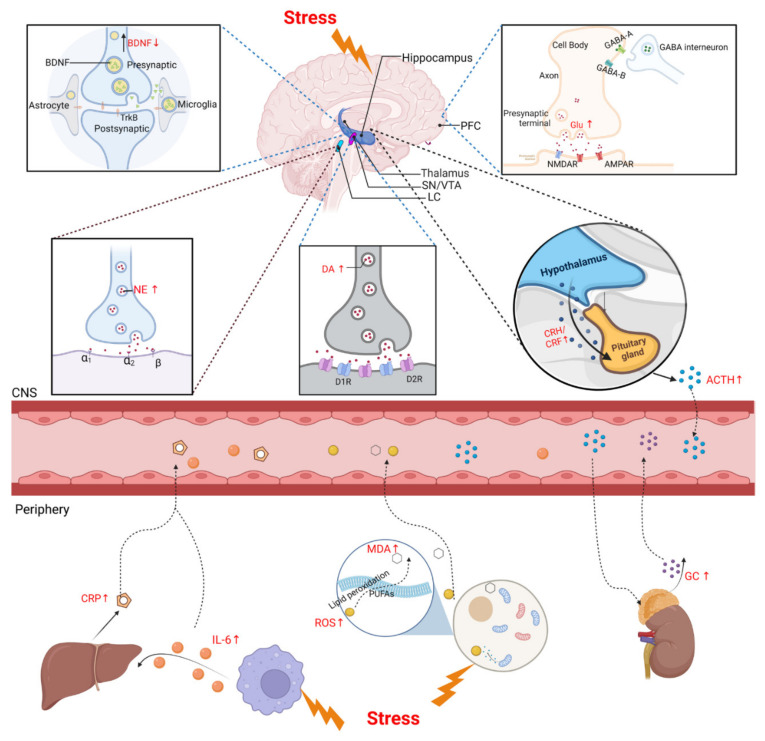
Biomarkers of CNS and periphery after stress. Upper panel shows HPA axis and LC-NE system are activated under stress. CRH and AVP, released by PVN through the pituitary portal system to the pituitary gland, act together on the pituitary gland to promote the release of ACTH through the circulatory system to the adrenal cortex, and then promote the synthesis and release of GCs. The LC-NE neurons can supply NE to modulate the stress response. BDNF is secreted by various CNS cells, such as neurons and astrocytes, and the level of BDNF protein decreases in the PFC after stress. DA is produced in the SN and the VTA of the midbrain. Exposure to acute stress shows an immediate increase in DA and Glu and a decrease in GABA. Lower panel shows the biomarkers of stress in the periphery. Acute stress induces an increase in IL-6, which stimulates the release of liver-derived CRP. Large amounts of ROS and MDA were also produced under stress, which then caused lipid peroxidation damage. ↑: upregulated; ↓: downregulated; BDNF: brain-derived neurotrophic factor; PFC: prefrontal cortex; SN: substantia nigra; VTA: ventral tegmental area; LC: locus coeruleus; CRH/CRF: corticotropin-releasing hormone/factor; ACTH: adrenocorticotropic hormone; Glu: glutamate; NE: norepinephrine; DA: dopamine; CNS: central nervous system; ROS: reactive oxygen species; MDA: malondialdehyde; PUFAs: polyunsaturated fatty acids; GC: glucocorticoid.

**Table 1 ijms-23-09519-t001:** The effects and mechanisms of stress on TBI.

Stress Models	TBI Models	Neurological Impairments and Mechanisms	References
RS/DEX #	mFPI	Hippocampus GR ↑; Hippocampus pro-BDNF ↓.	[139]
RS #	FPI	Plasma corticosterone level at the acute stage post-stressor initiation ↑.	[140]
CRS #	Moderate CCI	Motor deficits and cognitive impairment ↑; Anxiety-like behaviors and lesion volume ↑; BBB leakage, ERS, apoptosis, and autophagy ↑.	[141]
RUS #	r-mTBI	Traumatic memory impairments and anxiety-like and passive stress-coping behaviors ↓; Dendritic spine GluN2A/GluN2B ratio ↓; Pro-BDNF level in the hippocampus ↓; Astrogliosis in the corpus callosum ↑.	[142]
Foot shock #	rcTBI	Cognitive impairments ↑; Depression-like behaviors ↑.	[143]
Repeated immobilization and tail-shock stress #	mFPI	Anxiety and memory impairments ↑; Abnormal mitochondrial ETC complex and PDH enzyme expressions in hippocampus.	[144]
Repetitive unpredictable stressors #	bTBI	Anxiety-like behaviors ↑; Spatial memory impairments ↑; Corticosterone, CK-BB, NF-H, NSE, GFAP, and VEGF in the blood serum and the above protein levels in the hippocampus and the PFC ↑.	[145]
Social isolation #	CHI	Cognitive impairment ↑; Serum corticosterone ↑; Corticosterone ↑.	[146]
Social isolation #	A penetrating injury	Memory impairments ↑; Dark neurons and apoptotic cells in the hippocampal CA3 region ↑.	[147]
Forced wheel exercise #	FPI	Plasma corticosterone and ACTH ↑.	[148]
SF #	FPI	The upstream regulator NR3C1 that encodes GR ↓; Cortical microgliosis ↑; Pro-inflammatory glial signaling genes ↑; Neuronal activity in the hippocampus ↑; Neuronal activity in PVN ↓; Hippocampal-dependent cognition ↓.	[149]
PTSD (human) #	TBI	CRF ↓.	[150]
PTSD #	CHI	Behavioral impairments and neuroinflammation ↑; Microglia number in DG, CA1, and CA3 ↑.	[151]
Maternal separation @	FPI	Hippocampal-dependent learning deficits ↑; Cortical atrophy ↑; GCs in blood serum ↑.	[152]
Maternal separation @	Mild CCI	Executive function (-); Hippocampal IL-1β ↑; Plasma corticosterone level ↑.	[153]
Maternal separation @	CCI	Spatial learning and memory deficits ↓; Contralateral CA1 microglial activation ↑.	[154]
Foot shock @	CCI	sEPSC in lateral amygdala pyramidal-like neurons ↓.	[155]
Social defeat @	Mild CCI	Anxiety-like behaviors ↑; 5-HTP and 5-HIAA in hippocampus and amygdala ↑; DA in dorsal hippocampus ↑; NE in the amygdala ↑.	[156]

Note: #: post-injury stress; @: pre-injury stress; ↑: upregulated; ↓: downregulated; RS: restraint stress; DEX: dexamethasone; mFPI: mild fluid-percussion injury; GR: glucocorticoid receptor; BDNF: brain-derived neurotrophic factor; CRS: chronic restraint stress; CCI: controlled cortical injury; BBB: blood–brain barrier; ERS: endoplasmic reticulum stress; RUS: repeated unpredictable stress; r-mTBI: repetitive mild TBI; rcTBI: repetitive concussive TBI; ETC: electron transport chain; PDH: pyruvate dehydrogenase; bTBI: blast induced TBI; CK: creatine kinase; NF-H: neurofilaments-heavy; NSE: neuron-specific enolase; GFAP: glial fibrillary acidic protein; VEGF: vascular endothelial growth factor; PFC: prefrontal cortex.

## Data Availability

Data sharing is not applicable to this article as no datasets were generated or analyzed during the current study.

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
