# Peer review of "The Neurobiological Links between Stress and Traumatic Brain Injury: A Review of Research to Date"

_ijms, 2022, doi:10.3390/ijms23179519_

Round 1

Reviewer 1 Report

In this review, the authors investigated the effects of traumatic brain injury (TBI) and stress on neurological dysfunctions and different brain regions; also they explored the neurobiological links between stress and TBI,  summarized stress biomarkers, and probed the possible diagnostic and therapeutic significance on stress or stress plus mild and moderate TBI.

This manuscript is interesting; however it needs substantial improvements and corrections before publishing may be possible.

General points:

For better readability please add 2-3 Figures with appropriate Legend to your review.

Please add a List of abbreviations before References section to your manuscript.

Please do your List of References at the end of your manuscript according to IJMS.

 Special points:

Important, this manuscript should be substantially improved, i. e., by substantial references in the field.

Introduction

Lines 24-36: please add multiple references at the end of each of these sentences.

Main part

Lines 43-44: please add multiple references at the end of this sentence.

Lines 44-58: please add multiple references at the end of each of these sentences.

Lines 60-64: please correct the font.

Lines 76-84: please add multiple references at the end of each of these sentences.

Line 199: please add multiple references at the end of this sentence.

Lines 239-242: please add multiple references at the end of each of these sentences.

Line 374: please write out: ELS.

Lines 417-421: please add multiple references at the end of each of these sentences.

Lines 426-429: please add multiple references at the end of each of these sentences.

Lines 431-432: please add multiple references at the end of this sentence.

Lines 444-445: please add multiple references at the end of this sentence.

Lines 458-459: please add multiple references at the end of this sentence.

Lines 482-483: please correct the font.

Lines 486-487: please add multiple references at the end of this sentence.

Lines 512-514: please add multiple references at the end of each of these sentences.

Lines 526-527: please add multiple references at the end of this sentence.

Lines 550-552: please add multiple references at the end of this sentence.

Lines 554-559: please add multiple references at the end of each of these sentences.

Lines 565-569: please add multiple references at the end of each of these sentences.

Reviewer 2 Report

[IJMS] Manuscript ID: ijms-1849550 report

This is a wonderful effort of the authors to describe the multileveled effect of TBI together with stress.

Do the authors consider differences between localized and diffuse TBI ?  I mean, throughout the paper the neuroanatomical injuries due to the TBI discussed with the various subjects are not described.

The authors do not appear to differentiate between acute and chronic stress. Not always

Regarding RS, can the authors distinguish between the effects of the restraining itself and the stress ?

The authors consider two main consequences of TBI :

Attention deficit ( what kind of ?  lack of concentration ? )

Depression ( reduced interaction with the environment ? )

Just some thoughts here, since the paper wants to connect TBI and stress.  Let's say, in war TBI and stress (PTSD) most often are strongly connected.

Social and economical level may also correlated with TBI and Stress.  Meaning people in disadvantageous social and economical status are more vulnerable i.e. recover less well from TBI

The paper is neuropathologic rather than molecular biological, in particular because molecular biological mechanisms are not considered.

As the authors do not take TBI size into account, or TBI primary location to various effects described in the brain, it cannot be easily distilled from this paper what are the causes for the effects of TBI seen in the brain parts that are showing such effects. 

While the paper presents numerous and important details regarding TBI, also in connection with stress, there is no clear hypothesis how and by which mechanisms TBI may aggravate stress and vice versa. 

The conclusion appears to cover more than what is written in the body of the text.  Parts of it should be moved to the introduction. I have the impression that the authors have a bias toward glucortoids.  Of course there are many aspects underlying the reciprocal connections between stress and TBI.  The authors could rank order these aspects in a table.  For example how stress impacts gliosis, apoptosis, inflammation, cell damage, healing processes, restructuring of damaged tissue that are consequences of TBI.

Minor comments

Line 28 "exist" is the wrong word

Lines 29 to 33 are repetitive, instead of repetions more information can be presented.

Line 55 could you give other behavior emotional disorders ( affection, social interactions, aggression, things like that ) could you give motor dysfunction ?

In these lines, 53 to 55 the authors actually are not very precise in their terminology.  For example, cognitive impairments I would not call executive dysfunction.

84  (Attention disorder is one of the most common complaints of TBI survivors.)  This is something I didn't know.  

Line 95 " is " is not the right word

Line 161 "Aggragate´  do you mean "aggravate" ?

Line 371 "persist´ should be persistent.

Line 428 delete " have "

Line 463 " receptor receptor "  is this correct ?

Lin 483 " often lead "  should be " often leading "

Line 546 "neuron" is "neurons"

Line 554 "ROS is" should be "ROS are"

Line 581 "tolerate" is not the right word   "suffer" you mean, I guess.

Line 616 "Persist" you mean chronic.

Reviewer 3 Report

The authors present a manuscript about the relationship between neurobiological links between stress and traumatic brain injury. They have comprehensively reviewed both animal and human studies that contribute information about brain biomarkers, structure and function related to stress.  A few overall points need further clarification to enhance the scientific value of the manuscript:

-It would be helpful to explain how a traumatic brain injury can change the biophysiology of brain function. This is mentioned briefly on page 6 but newer literature in this area further describes that an impact to the brain does to change brain functioning. 

-Not sure how "pre-injury" stress is related to the overall issues of TBI. Not everyone experiences preinjury stress. Although this is interesting, it would be better to focus on the changes that the TBI injury brings to brain function.

-Experiencing a TBI itself can be stressful...depending on the mechanism of injury ( i.e. assault compared to falls or sports related injuries.). Based on your knowledge of brain physiology, how does experiencing the TBI injury contribute to this aspect?

-There is more recent literature on TBI biomarkers that needs to be included. For example, work has been done on TBI and Autonomic nervous system and TBI and blood biomarkers

-TBI can be a complex injury just in itself. It would be best to focus the paper simply on the TBI-injury itself rather than include preinjury stress which complicates the understanding of TBI and stress. 

Round 2

Reviewer 1 Report

Thank you for your corrections. Unfortunately, this manuscript needs some corrections before publishing may be possible.   Please correct the References list according to “IJMS”.   Please add a Legend with description to Figure 1.

Reviewer 2 Report

Your hard work and strong interest in the subject is very much appreciated.